# Advancing Cancer Therapy: The Role of KIF20A as a Target for Inhibitor Development and Immunotherapy

**DOI:** 10.3390/cancers16172958

**Published:** 2024-08-24

**Authors:** Dong Oh Moon

**Affiliations:** Department of Biology Education, Daegu University, 201, Daegudae-ro, Gyeongsan-si 38453, Gyeongsangbuk-do, Republic of Korea; domoon@daegu.ac.kr

**Keywords:** KIF20A, peptide vaccine, cancer, inhibitor

## Abstract

**Simple Summary:**

This review focuses on the current advancements in the development of inhibitors and the application of kinesin family member 20A (KIF20A) as a cancer immunotherapy target. KIF20A, a mitotic kinesin overexpressed in various cancers, plays a pivotal role in cell division and is increasingly recognized as a crucial factor in tumor progression and a potential target for novel cancer treatments.

**Abstract:**

The analysis begins with a detailed examination of the gene expression and protein structure of KIF20A, highlighting its interaction with critical cellular components that influence key processes such as Golgi membrane transport and mitotic spindle assembly. The primary focus is on the development of specific KIF20A inhibitors, detailing their roles and the challenges encountered in enhancing their efficacy, such as achieving specificity, overcoming tumor resistance, and optimizing delivery systems. Additionally, it delves into the prognostic value of KIF20A across multiple cancer types, emphasizing its role as a novel tumor-associated antigen, which lays the groundwork for the development of targeted peptide vaccines. The therapeutic efficacy of these vaccines as demonstrated in recent clinical trials is discussed. Future directions are proposed, including the integration of precision medicine strategies to personalize treatments and the use of combination therapies to improve outcomes. By concentrating on the significant potential of KIF20A as both a direct target for inhibitors and an antigen in cancer vaccines, this review sets a foundation for future research aimed at harnessing KIF20A for effective cancer treatment.

## 1. Introduction

In oncology, the identification and understanding of key molecular drivers are essential for developing diagnostic, prognostic, and therapeutic strategies. KIF20A, a mitotic kinesin, plays a significant role in cell division and intracellular transport and is notably overexpressed in various cancers, making it a potential target for cancer therapy [1,2,3,4]. This review focuses primarily on the expression of KIF20A in tumors, its protein structure, current developments in its inhibitors, and its application in cancer immunotherapy as a novel tumor-associated antigen (TAA).

The recent advances in molecular biology have emphasized KIF20A’s critical functions in cellular processes and its abnormal expression in different cancers. Its interaction with transcription factors like Forkhead Box M1 (FOXM1) and influences from the tumor microenvironment have been documented, affecting its activity in cancer cells [5,6]. The structural domains of KIF20A facilitate its roles in microtubule dynamics and vesicle transport, which are pivotal in cellular homeostasis and can contribute to cancer development when dysregulated [7].

Clinically, KIF20A is recognized for its prognostic value in melanoma, breast, bladder, cervical, ovarian, and gastric cancers, serving as a promising biomarker for tracking cancer progression. The exploration of KIF20A has also advanced targeted therapy options, particularly in manipulating pathways such as JAK/STAT3 signaling, which are crucial for innovative treatments in colorectal cancer and other malignancies [8,9].

Advancements in cancer immunotherapy have leveraged KIF20A as a novel antigen to develop peptide vaccines, enhancing the immune system’s ability to target cancer cells [10]. The clinical trials targeting KIF20A have shown promising results, affirming the feasibility and effectiveness of KIF20A-based immunotherapeutic approaches.

This review aims to consolidate the existing knowledge on KIF20A within the cancer research field, detailing its crucial role in tumorigenesis and its potential to enhance the efficacy of cancer diagnostics, prognostics, and treatments. By focusing on the molecular behavior and clinical implications of KIF20A across various cancers, this review underscores its importance in the era of precision medicine, paving the way for more targeted and effective cancer therapies.

## 2. KIF20A Expression and Protein Structure

The KIF20A gene is located on chromosome 5, spanning positions 138,178,719 to 138,187,723 based on the GRCh38/hg38 reference genome. It encompasses a total length of 9005 bases and consists of 19 exons [11,12].

The activation of glioma-associated oncogene family zinc finger 2 (Gli2), a central figure in Hedgehog (Hh) signaling pivotal for hepatocellular carcinoma (HCC) development, has been found to indirectly stimulate the expression of KIF20A through the activation of FOXM1. This Gli2-KIF20A axis emerges as a crucial mechanism in the proliferation and survival of HCC cells, highlighting a potential therapeutic target within the Hh signaling pathway [13,14,15]. At the promoter region of KIF20A, a pivotal forkhead responsive element (FHRE) is located at +80 base pairs from the transcription start site, serving as a crucial docking point for the transcription factor FOXM1. The attachment of FOXM1 to the FHRE significantly increases the transcriptional activity of KIF20A, playing a key role in regulating the cell cycle. Importantly, in cancer cells, FOXM1 is often overexpressed [16,17], enhancing its ability to drive the expression of genes like KIF20A that are critical for cell proliferation and tumor progression. Understanding the dynamic between FOXM1 and FHRE within the KIF20A promoter could pave the way for novel cancer treatment modalities, particularly where the dysregulation of KIF20A contributes to disease progression.

Furthermore, lactate, which frequently accumulates in the tumor microenvironment, has been identified as a modulator of cell motility in cancer cells [18,19]. Computational analyses suggest kinesin family members, including KIF20A, are instrumental in this process. It has been elucidated that lactate upregulates E2F1, which then orchestrates the microtubule dynamics essential for lactate-driven cell motility, employing kinesin proteins as its agents [20]. Therefore, the lactate-induced E2F1 activation, with its downstream regulation of KIF20A, contributes to the carcinogenic cascade, fostering the migration and metastasis of cancer cells and delineating a lactate-E2F1-KIF20A pathway integral to cancer progression.

The KIF20A protein is structured into three distinct regions: an N-terminal motor domain (amino acids (aa) 56 to 505), a central helical domain (aa 519 to 821), and a C-terminal tail domain (aa 821 to 890), each with specific roles and characteristics that are crucial for the protein’s function in cellular mechanisms [21]. The motor domain at the N-terminus is defined by a classic kinesin architecture, yet it stands out due to its distinctive structural elements, such as the notably open nucleotide-binding site (NBS) and a long insertion in loop L6 (Gly192-Asp295).

The NBS of KIF20A is unusually accessible to solvents, which is not common among kinesins [22,23]. Studies using techniques like cryo-electron microscopy and X-ray diffraction show that this open configuration remains largely unchanged even when microtubules are present [23,24]. This suggests that KIF20A’s NBS naturally stays open, making the active site more accessible. This feature is important for KIF20A’s interaction with ATP and could influence its mechanical properties. The ability of KIF20A to bind and break down ATP, necessary for its movement, is aided by the open NBS which facilitates easier access to nucleotides. Moreover, the openness of the NBS is key to understanding how KIF20A functions as a motor protein, using ATP to move along microtubules towards their plus end.

Loop L6 in KIF20A contains an unusually long insertion of residues that significantly differs from that observed in other kinesins like Kin-1. This long insertion contributes to the unique structural and functional aspects of KIF20A. Structurally, the L6 loop with its extended insertion impacts the overall conformation of the motor domain, particularly influencing its interactions with other parts of the protein and the microtubules [22,25]. Functionally, this extended loop plays a critical role in the mechanochemistry of KIF20A, affecting how it interacts with microtubules and possibly its processivity and speed as a motor protein [22]. The specifics of these insertions, including their secondary structure elements such as β-hairpins and helices, are key to understanding the distinctive kinetic behavior and regulatory mechanisms of KIF20A compared to other kinesin family members.

The central helical domain, though not extensively described, is presumed to play a significant role in dimer formation and interaction with binding partners. This segment’s functionality is crucial for KIF20A, facilitating dimerization for its operational effectiveness. It likely participates in binding activities with Rab6 and myosin II, underscoring its importance in vesicle transportation and cell division, as evidenced by its role in the detachment of vesicles from the Golgi apparatus [26].

The domain at the C-terminus is crucial for vesicle transport and cargo interaction. It enables KIF20A to attach to and ferry vesicular cargo within cells [27]. This domain is critical for accurately positioning mitotic regulators and various essential molecules during cell division, contributing to cytokinesis and potentially influencing the development of certain cancers. These combined characteristics of KIF20A’s domains elucidate its multifaceted roles in cellular processes, highlighting its potential as a therapeutic target, especially considering its upregulation in various cancers. The gene expression and protein structure of KIF20A are depicted in Figure 1.

## 3. KIF20A Expression in Cancer: Implications for Prognosis and Therapeutic Strategies

The expression of KIF20A has been highlighted across various studies as a pivotal factor in the prognosis and progression of numerous cancers. Jie Jiang et al. have pinpointed KIF20A among five key genes with elevated expression correlating to adverse outcomes in melanoma, suggesting its utility as a biomarker for diagnosis, treatment, and prognosis [28]. Masako Nakamura’s research elucidates KIF20A’s pronounced expression in different breast cancer subtypes compared to normal tissue, identifying it as an independent marker of prognosis and a potential target for therapy, with its suppression resulting in a halted cell cycle and cancer cell death [29]. Tianyu Shen’s findings reveal a significant upregulation of KIF20A in bladder cancer, associated with increased proliferation and metastasis, marking it as a crucial prognostic factor and therapeutic target [3]. Weijing Zhang and the team have shown that elevated KIF20A expression in early-stage cervical squamous cell carcinoma is linked to HPV infection, advanced stage, and heightened risk of recurrence and metastasis, underscoring its role as an independent prognostic indicator and therapeutic target [30]. Han Li’s study connects overexpression of KIF20A in epithelial ovarian cancer with advanced stages, metastasis, recurrence, and chemoresistance, establishing it as a significant independent risk marker and promising prognostic biomarker [31]. Yi Sheng and collaborators have demonstrated that KIF20A’s increased expression in gastric cancer correlates with enhanced cell proliferation, poor overall survival, and higher histological grades, highlighting its oncogenic role and its potential as a prognostic and therapeutic target [32]. Jia Duan et al. found that positive KIF20A expression in glioma is significantly associated with higher Ki67 protein expression, advanced tumor grades, and diminished survival, positioning KIF20A as an independent prognostic factor and suggesting its contribution to tumor proliferation, invasion, and possible chemotherapy resistance [33]. Runxiang Qiu and associates provided evidence that targeting KIF20A in medulloblastoma leads to premature cell cycle exit and enhanced neuronal differentiation, suppressing tumor growth and proposing a novel anti-proliferative treatment strategy by affecting cell fate determination [34].

TNMPlot, a platform rich in RNA-Seq and gene chip data, offers advanced statistical analysis tools like *p*-values and fold change indicators for deep gene significance insights. Its capacity to rapidly assess KIF20A expression patterns across various cancers, combined with a user-friendly interface designed for selecting specific cancers and comparing tumor versus normal samples, establishes it as a key resource for cancer research gene expression profiling [35,36]. In an extensive review of RNA-Seq and gene chip data sourced from TNMPlot, KIF20A expression was substantially elevated in several cancer types relative to normal tissue, with RNA-Seq data demonstrating the most significant increases in cholangiocarcinoma, showing a mean fold change of 67.07 and a median of 42.67, and liver hepatocellular carcinoma, with a mean fold change of 20.78 and a median of 16.50. Bladder urothelial carcinoma also showed notable upregulation with a median fold change of 19.00. Other cancers, such as cervical squamous cell carcinoma and endocervical adenocarcinoma, revealed a striking median fold change of 74.08, indicating a profound differential in KIF20A expression that may correlate with disease severity or prognosis. The gene chip data corroborated these findings, with esophageal cancer showing a pronounced mean fold change of 4.85 and ovarian cancer displaying an increase with a mean fold change of 3.41. These data points, indicating the heightened presence of KIF20A in cancerous versus normal tissues, highlight its potential role in oncogenesis and its utility as a diagnostic and prognostic biomarker, suggesting further exploration into KIF20A’s function in cancer biology and therapy could be fruitful. Table 1 presents a summary of KIF20A expression across different cancer types, as determined by TNMPlot analysis.

## 4. Multiple Roles of KIF20A

The molecular intricacies of cellular functions and their deviations in pathological conditions, particularly in cancer, are areas of intense research and fundamental interest. The KIF20A protein, a key player in intracellular transport mechanisms, has emerged as a focal point in understanding these complex biological processes. This review delves into the multifaceted roles of KIF20A, ranging from its involvement in the transport of Golgi membranes and associated vesicles to its critical functions in mitosis, including the recruitment of Polo-like kinase 1 (PLK1) to the central spindle, the coordination with the chromosome passenger complex (CPC) during cytokinesis, and its role in activating the Janus kinase (JAK)-signal transducer and activator of transcription (STAT) 3 pathway, which has significant implications for cancer progression. The content regarding the transport of Golgi membranes and associated vesicles is depicted in Figure 2.

### 4.1. Transport of Golgi Membranes and Associated Vesicles

KIF20A is directly involved in the fission of RAB6-positive vesicles at the Golgi complex. Studies have shown that KIF20A, along with Myosin II, contributes to the formation of what are termed Golgi fission hotspots [37]. These are specific areas on the Golgi membrane where vesicles exit into the cytoplasm. The fission process is facilitated by KIF20A’s interaction with Myosin II, enhancing vesicle trafficking along microtubules. This interaction is crucial for maintaining the structural integrity and functional capacity of the Golgi in secretion processes [37,38,39]. The transport of RAB6-positive vesicles along microtubules is a critical aspect of cellular logistics, allowing for the distribution of molecular cargoes throughout the cell. KIF20A’s role extends beyond vesicle fission, participating in the microtubule-dependent movement of these vesicles. This is facilitated by KIF20A’s motor activity, which is essential not only for moving vesicles along microtubules but also for organizing these cellular structures to optimize transport efficiency [37,40]. The crystal structure of the RAB6 complex reveals that the interaction is mediated through the 603–645 region of KIF20A, which forms a dimerized right-handed coiled-coil. This dimer is stabilized by an inter-helical cysteine bridge, enhancing the binding specificity. RAB6 molecules attach to opposite sides of this dimeric structure. Specific amino acid residues in KIF20A, particularly K629 and S630, are critical for forming polar contacts with RAB6, facilitating a robust interaction that is essential for the proper handling and direction of RAB6-positive vesicles towards their intended cellular locations [26]. This detailed interaction highlights the sophisticated coordination between KIF20A and RAB6 in vesicle trafficking processes [38]. The cryo-electron microscopy (cryo-EM) study of KIF20A bound to microtubules reveals two structural classes, with the majority of particles showing well-defined microtubule-binding loops. These loops, particularly L2 and L12, are longer in KIF20A than in other kinesins, facilitating its interaction with microtubules. In the dominant class, the loops L9 and L11 form a stable salt bridge that is crucial for ATP hydrolysis, a key process in KIF20A’s motor function along microtubules [7]. This interaction enables KIF20A to effectively transport cellular components along microtubules, highlighting its role as a dynamic motor protein in cellular logistics. Effective and precise control of vesicle trafficking is especially critical in cancer cells, where enhanced secretion processes support rapid cell proliferation and tumor growth. Disruptions in vesicle trafficking can result in aberrant signaling and cellular dysfunction, contributing to cancer progression and metastasis. Therefore, understanding and potentially targeting KIF20A’s role in these processes could be crucial for developing new therapeutic strategies against cancer.

### 4.2. Recruitment of PLK1 to the Central Spindle

In the cellular process of mitosis, a critical mechanism involves the phosphorylation of KIF20A by PLK1, which significantly contributes to the recruitment and proper localization of PLK1 to the central spindle, a key area for chromosome segregation and cytokinesis [27,41]. PLK1, a serine/threonine-protein kinase, is essential for several mitotic events, including the precise segregation of chromosomes and the extension of the anaphase B spindle, that are vital for the successful completion of cell division.

However, the precise localization of PLK1 to the spindle midzone is a complex process that requires specific interactions with other proteins, notably KIF20A. KIF20A acts as a scaffold that facilitates the assembly of PLK1 at the central spindle. This interaction is mediated through a phosphorylation-dependent mechanism, where KIF20A undergoes phosphorylation at a specific sequence, enhancing its binding affinity to PLK1 [27]. This phosphorylation site on KIF20A, identified as EHS528LQV, plays a crucial role in enabling PLK1 to recognize and bind to KIF20A through its polo-box domain, a specialized region within PLK1 designed to interact with phosphorylated targets [21,42]. PLK1 then adds phosphate groups to KIF20A, resulting in their complex formation. During the transition from metaphase to anaphase, KIF20A plays a crucial role in transporting PLK1 from the centrosome to the spindle midzone.

The dynamics of this interaction highlight a finely tuned regulatory system where PLK1 not only targets KIF20A for phosphorylation more preferentially than other mitotic proteins such as MKlp1 but also controls KIF20A’s ability to bind to microtubules [41]. This regulation is crucial for maintaining the integrity and functionality of the mitotic spindle. Furthermore, this interaction underscores a reciprocal regulation where the activity of microtubules can influence the efficiency and specificity of PLK1’s phosphorylation of KIF20A. Such a feedback loop suggests that the spatial and temporal coordination of PLK1 and KIF20A activities is intricately linked to the dynamic state of microtubules during mitosis, reflecting a complex interplay of signaling and structural components critical for cell division.

### 4.3. CPC-Mediated Cytokinesis

During cytokinesis, the phase of cell division where the cell physically splits into two daughter cells, the dynamic interaction between KIF20A and the CPC is critical for ensuring the fidelity and accuracy of this process [21,43]. The CPC is a molecular ensemble composed of four core components: Aurora B kinase, inner centromere protein (INCENP), Survivin, and Borealin [44,45]. Each of these components plays a unique role in mitosis, with Aurora B kinase acting as the catalytic center of the complex [46]. This kinase is responsible for phosphorylating various substrates involved in chromosome alignment, spindle assembly checkpoint, and cytokinesis [47]. INCENP serves as a scaffolding protein that binds directly to Aurora B, enhancing its kinase activity and directing it to its substrates [48]. Survivin and Borealin, on the other hand, contribute to the stability and localization of the CPC, facilitating its attachment to centromeres and the central spindle, areas critical for its function [49,50].

KIF20A is integral to this process by ensuring the proper localization of the CPC during cytokinesis [27]. It accomplishes this through its motor activity, which allows it to traverse microtubules, positioning itself and the CPC at the spindle midzone. The spindle midzone, a region rich in overlapping microtubules formed during anaphase, is pivotal for the recruitment and function of proteins necessary for cytokinesis.

The interaction between KIF20A and the CPC is a delicate balance of molecular signaling and localization, coordinated by phosphorylation events. KIF20A’s motor activity and its interaction with microtubules are regulated by its phosphorylation state, which, in turn, is influenced by cell cycle regulators such as cyclin-dependent kinases (Cdks) [27]. During early mitosis, KIF20A is phosphorylated and inactive, preventing premature recruitment of the CPC to the central spindle. As the cell progresses through mitosis, dephosphorylation of KIF20A activates it, allowing for its movement to the spindle midzone, where it plays a crucial role in stabilizing microtubules. This stabilization is vital for the recruitment of the CPC, which then executes its function in ensuring the chromosomes are correctly aligned and segregated, and the cell is ready to divide.

### 4.4. Activation of the JAK/STAT3 Pathway

The JAK/STAT3 pathway is a critical signaling mechanism within cells, playing a pivotal role in various biological processes including cell growth, differentiation, and immune response regulation [51,52]. In the context of cancer, this pathway assumes an even more sinister importance, being integral to the growth, survival, invasiveness, and metastatic potential of cancer cells. Its aberrant activation is frequently observed in many types of cancer, including colorectal cancer, where it contributes to the malignant phenotype of the cancer cells. The JAK/STAT3 pathway’s role in mediating the effects of KIF20A highlights its importance in colorectal cancer progression [53]. This pathway, when activated, can lead to the transcription of genes that promote cancer cell proliferation, survival, and invasion. The activation of this pathway typically begins with extracellular signals, such as cytokines and growth factors, binding to receptors on the cell surface, which leads to the phosphorylation of JAK kinases. This phosphorylation event subsequently recruits and activates STAT3 proteins through phosphorylation. Once activated, STAT3 proteins dimerize and translocate into the nucleus, where they serve as transcription factors. The genes regulated by STAT3 are often involved in critical processes like cell cycle progression, inhibition of apoptosis, and angiogenesis, all of which contribute to the malignancy of colorectal cancer cells. Activated by tyrosine kinases such as JAK2, STAT3 dimerizes and moves to the nucleus to regulate gene expression. STAT3 activation has been closely associated with chemoresistance in several cancers. A study by Man Xiong et al. showed that the increased expression of KIF20A can activate the JAK2/STAT3 signaling pathway by promoting the phosphorylation of JAK2 and STAT3 [53]. Understanding this interaction not only sheds light on the molecular underpinnings of colorectal cancer but also opens the door for developing targeted therapies that could inhibit the JAK/STAT3 pathway, KIF20A expression, or both, offering hope for more effective treatments for this challenging disease. The various intracellular functions of KIF20A are illustrated in Figure 2.

## 5. Types and Functions of KIF20A Inhibitors

The first inhibitor of KIF20A, named Paprotrain, was developed by Sergey Tcherniuk et al. [54]. The chemical structure of Paprotrain is (Z)-2-(1H-indol-3-yl)-3-(pyridin-3-yl)acrylonitrile. This compound acts as an uncompetitive inhibitor with respect to ATP and noncompetitively with microtubules (MTs), indicating that it may bind at a different site from ATP or MTs, or it induces a structural change in the protein after binding, thus inhibiting ATPase activity. The inhibition efficiency shows a consistent inhibitory effect irrespective of substrate concentration, with Ki values approximately 3.4 μM for ATP and 1.6 μM for MTs. Cell permeability experiments demonstrated that Paprotrain can penetrate cell membranes when tested in HeLa cells at concentrations ranging from 10 μM to 50 μM.

Next, we examine the oncogenic impact of Paprotrain, a targeted KIF20A inhibitor, on diverse cancer cell models. Formulated by Sergey Tcherniuk and his team, this agent specifically disrupts the trafficking of chromosome passenger proteins, arrests cytokinesis, and results in the emergence of binucleated cells, all while selectively sparing non-targeted kinesin proteins [54].

Ingrid E. Adriaans et al. provided valuable insight demonstrating that Paprotrain notably hinders the motor activity and transport efficiency of KIF20A, critically impacting the localization and mobility of CPC in anaphase without affecting related kinesin-6 family members [55]. This insight underscores the potential of Paprotrain to selectively disrupt specific cellular functions, offering a targeted approach in cancer therapy by interfering with mitotic processes while preserving other crucial cellular mechanisms.

Valeria A. Copello and Kerry L. Burnstein conducted significant research showing that Paprotrain plays a crucial role in mitigating the progression of castration-resistant prostate cancer [56]. By effectively inhibiting KIF20A, Paprotrain disrupts the autocrine activation mechanisms of the androgen receptor (AR), a key driver of cancer proliferation in prostate cells that have become resistant to traditional androgen-deprivation therapies. This inhibition results in a marked decrease in the aggressive behavior and growth of castration-resistant prostate cancer cells, highlighting the potential of Paprotrain as a promising therapeutic agent in the management and treatment of advanced prostate cancer.

Julian Kositza and colleagues conducted a study that revealed the effectiveness of combining Paprotrain, a specific inhibitor of KIF20A, with palbociclib, a well-known CDK4/6 inhibitor, in treating bladder cancer. Their research showed that this combination leads to a synergistic effect, significantly reducing the viability of cancer cells more than either drug could achieve alone. This enhanced suppression of cell growth in bladder cancer cell lines suggests a highly potent improvement in therapeutic efficacy, making it a promising strategy for more effective cancer treatment [57].

Morgan S. Schrock and colleagues conducted a detailed study demonstrating that Paprotrain plays a critical role in causing mitotic arrest. Their research revealed that this compound significantly disrupts the process of chromosome alignment, leading to pronounced chromosomal instability. Specifically, the inhibition of KIF20A by Paprotrain was shown to interfere with the proper congression of chromosomes during prometaphase. This disruption was further linked to errors in kinetochore–microtubule attachments, exacerbated by increased activity and phosphorylation of Aurora kinases A and B, and their downstream target, HEC1, at Ser55. This cascade of molecular events confirms that Paprotrain’s mechanism of action includes both hindering chromosome congression and altering the regulatory landscape of key mitotic kinases, thereby underscoring its profound impact on cellular division and stability [58].

Compound 9a, a derivative developed by the same group that created Paprotrain, was synthesized as part of a series targeting KIF20A [59]. This synthesis utilized a Knœvenagel condensation of indol-3-ylacetonitrile with various benzaldehydes to introduce different aromatic substituents into the structure. Specifically, 9a was produced by coupling pyridine-3-carbaldehyde with appropriate arylacetonitriles, followed by reactions under Suzuki conditions to add further groups. This process included selective bromination and subsequent coupling steps to enhance the inhibitory activity of the indole ring by adding functional groups. The introduction of a methoxy group at the 5-position of the indole ring in compound 9a significantly enhanced its inhibition against human tumor cell lines and improved its biochemical potency, achieving IC50 values of 1.2 µM for 70% inhibition of basal KIF20A ATPase activity and 0.23 µM for 90% inhibition of microtubule-stimulated activity, both markedly better than the original Paprotrain. Furthermore, 9a did not influence MT dynamics, which contrasts with many other compounds that affect tubulin polymerization and depolymerization. In cytotoxicity tests, 9a demonstrated a significant inhibition of KB tumor cell growth, particularly at a concentration of 10 µM where it showed 57% growth inhibition. These findings suggest that 9a, with its enhanced potency and specificity, represents a promising candidate for clinical applications in cancer therapy, particularly due to its potent effects in various human cancer cell lines, including a notable impact on MIA-PaCa-2 human pancreatic cancer cells. This positions compound 9a as a potent inhibitor with the potential for development into an effective therapeutic agent in oncology.

In a recent study, H Ferrero and his research team identified BKS0349 as an effective antagonist of KIF20A. Their findings revealed that this compound significantly diminishes endometriotic lesions within a xenograft mouse model by both promoting apoptosis and curbing cell proliferation. These actions suggest that BKS0349 holds great promise as a new therapeutic option for managing endometriosis. It is important to note that the molecular structure of BKS0349 remains unpublished, underscoring the novelty and proprietary nature of this potential treatment [60]. The functions of KIF20A inhibitors in cancer treatment are detailed in Table 2.

The previous studies lacked molecular docking results; therefore, for this review, the compound structures of Paprotrain, Compound 9a and the crystal structures of the KIF20A protein were sourced from the PubChem (https://pubchem.ncbi.nlm.nih.gov/) and RCSB (https://www.rcsb.org/) databases, respectively, accessed on 5 May 2024 Molecular docking was then performed using CB-Dock (http://cao.labshare.cn/cb-dock/) to explore the interactions between inhibitors and the target KIF20A network, accessed on 6 May 2024.

ATP, the natural ligand for KIF20A, exhibits the strongest binding affinity with a Vina score of −7.7. It fits precisely within the ATP-binding site, occupying a relatively small cavity of 287 cubic angstroms at coordinates (−15, 4, 21). This ideal binding is crucial for ATP’s role in powering KIF20A’s motor activity.

Paprotrain, although showing a slightly less robust interaction with a Vina score of −7.6, occupies a larger cavity of 1109 cubic angstroms at coordinates (−13, 13, 29). It engages with KIF20A primarily through hydrogen bonds with Tyr339, Pro338, Val466, Glu402, and Arg474, π-π stacking with Phe470, ionic interaction with Glu402, and hydrophobic bonds with Trp345, Leu343, Phe305, and Val340. Despite the similarity in Vina scores with ATP, Paprotrain’s mechanism as an uncompetitive inhibitor involves binding to a state of the enzyme that is altered by ATP binding, suggesting it does not directly compete with ATP at the binding site but rather influences the enzyme function after ATP binding, potentially causing allosteric effects or modifications to the enzyme’s conformation.

Compound 9a, an analog of Paprotrain, exhibits a pattern of interaction characterized by a Vina score of −6.8 and binds within an even larger cavity of 1438 cubic angstroms, centered at (−13, 6, 33). The dimensions of this cavity (21, 28, 21) and its binding interactions through hydrogen bonds with Glu306, π-π stacking with Tyr313, and hydrophobic bonds with Glu317, Leu315, Ala368, and Tyr308 suggest a distinct mode of inhibition. Despite being an analog meant to enhance Paprotrain’s efficacy, the lower Vina score of Compound 9a compared to Paprotrain could be attributed to the less optimal interaction dynamics within this enlarged cavity, which might influence the overall stability and efficacy of its binding. The comparative docking analysis of ATP, Paprotrain, and Compound 9a on the N-terminal motor domain of KIF20A is depicted in Figure 3.

These findings highlight the intricate nature of drug interactions with KIF20A, a key player in cancer progression. Both Paprotrain and Compound 9a exhibit significant inhibitory properties, yet the nuances in their binding efficiency and characteristics are crucial for advancing our understanding. This knowledge is essential for refining these compounds to maximize their therapeutic potential, especially in targeting cancers associated with overactive KIF20A, which could lead to more effective treatments for such malignancies. 

The study of KIF20A inhibition and its effects on normal tissues remains insufficiently explored. Recent findings indicate that KIF20A suppression leads to early cell cycle exit and accelerated neuronal differentiation in both normal cerebellar granule neuron progenitors (GNPs) and tumor-initiating GNPs [34]. These results suggest that KIF20A plays a crucial role not only in tumor progression but also in the normal development of cerebellar neurons. Furthermore, an analysis of breast cancer subtypes revealed that KIF20A was expressed in 195 (75.9%) tumor samples, while it was rarely detected in normal breast tissues [29]. This indicates that KIF20A inhibition may have minimal impact on certain normal tissues. Nevertheless, comprehensive studies are required to better understand the broader implications of KIF20A inhibition on normal cells, as well as to determine its therapeutic potential across different cancer types. Further research is essential to evaluate the selective targeting of KIF20A in cancer treatment while minimizing adverse effects on normal tissues.

## 6. The Emergence of KIF20A as a Pivotal Target in Cancer Immunotherapy

The orchestration of effective anti-tumor immunity necessitates a sophisticated interplay among diverse immune cells and the tumor microenvironment (TME), emphasizing the complexity crucial to the success of cancer immunotherapy. Innate immune components such as NK cells, neutrophils, and macrophages play foundational roles, being among the first responders to tumor cells. Dendritic cells (DCs) are pivotal within this ecosystem, enhancing antigen presentation and T cell activation by capturing and presenting tumor antigens [61,62]. This dynamic is further enriched by immunogenic cell death, which not only releases tumor antigens but also induces DC maturation and subsequent T-cell-mediated immune responses, thereby fostering tumor immunity [63,64]. The support of CD4+ T cells for CD8+ T cell activation, along with the emerging concept of T cell stemness, highlights the intricacies of the immune response to tumors. Cancer vaccines are strategically designed to stimulate adaptive immunity against tumors by combining selected tumor antigens with adjuvants to activate DCs, aiming for tumor regression and potential disease eradication. Despite the promise of this strategy, challenges such as intrinsic tumor resistance and systemic immunosuppression persist, as evidenced by the varied success rates of past cancer vaccine trials [65,66]. The FDA’s approval of sipuleucel-T, a DC-based vaccine, along with the potential of immune checkpoint inhibitors (ICI) to enhance vaccine efficacy, underscores the ongoing refinement of immunotherapeutic strategies.

The efficacy of therapeutic cancer vaccines significantly depends on the choice of antigens, including both tumor-specific antigens (TSAs) and tumor-associated antigens (TAAs) [67,68]. TSAs, unique to cancer cells, offer specificity in targeting, minimizing the risk of attacking normal cells, while TAAs, although more abundant and expressed across various tumors, pose a potential risk of autoimmunity due to their presence in both normal and tumor cells. Despite this risk, TAAs are advantageous due to their wide recognizability by the immune system, which can be harnessed to broaden the immune response’s target spectrum. This potentially overcomes limitations associated with the availability and variability of TSAs across different tumors. Recent trends towards focusing on neoantigens and somatic mutations considered TSAs for their tumor-specific nature emphasize leveraging high specificity without autoimmunity risk, ensuring robust T cell responses without germ line tolerance [69,70]. The tumor mutational burden (TMB) plays a crucial role in neoantigen vaccine success, influencing neoantigen presentation and response to ICIs [71]. The neoantigens’ quality, including their novelty and major histocompatibility complex (MHC) presentation capability, underscores the importance of precision in neoantigen prediction.

In the milieu of cancer, TSAs and TAAs are proteins that typically arise from mutations or are aberrantly overexpressed [72]. These proteins are processed inside cancer cells for presentation by the MHC Class I pathway [73]. Proteasomes, including specialized immunoproteasomes, degrade these proteins into peptide fragments, which are then transported into the endoplasmic reticulum (ER) by the transporter associated with antigen processing (TAP) [74]. In the ER, these peptides encounter MHC Class I molecules. These molecules are held by chaperones such as calreticulin and ERp57, and the ‘peptide editor’ Tapasin, which facilitate the peptide loading. An enzyme called ERAP1 trims these peptides to a length that fits MHC Class I molecules precisely. Once the peptides are bound, the peptide-MHC Class I complexes are transported to the cell surface.

Mechanistically, TSAs and TAAs are recognized by DCs, processed, and presented on their surface within MHC molecules [75]. While KIF20A itself is not released from cells, the protein can still be involved in the immune surveillance process. Dendritic cells are capable of capturing cellular debris from tumors, which may include internal proteins like KIF20A, through cross-presentation. This mechanism allows dendritic cells to process and present both extracellular and intracellular antigens on MHC class I and II molecules. This antigen presentation is key to T cell priming, where T cell receptors (TCRs) recognize the antigen-MHC complex, leading to T cell activation. For TSAs, this response is highly specific, targeting antigens unique to tumor cells, thereby minimizing off-target effects. In contrast, the response to TAAs must be carefully managed to avoid potential autoimmunity due to these antigens’ presence in normal tissues. The activation of CD8+ T cells follows their recognition of specific antigens presented on tumor cells’ MHC class I molecules. Once activated, these CD8+ T cells proliferate and differentiate into effector cells, capable of directly killing tumor cells. This cytotoxic action is primarily mediated through the release of perforin and granzymes, which induce apoptosis in target cells [76], and through the expression of Fas ligand, which binds to Fas on tumor cells, also triggering apoptosis [77]. Additionally, activated CD8+ T cells can secrete pro-inflammatory cytokines like IFN-γ, further enhancing the immune response against the tumor by upregulating MHC expression and antigen presentation, thereby increasing tumor cell visibility to the immune system [78]. This comprehensive approach to antigen selection, integrating both TSAs and TAAs, aims to maximize vaccine efficacy by catering to cancer’s heterogeneity and the complexity of its interaction with the immune system. By understanding and leveraging the mechanisms by which TSAs and TAAs are presented on MHC molecules and how activated CD8+ T cells mediate tumor cell destruction, cancer immunotherapy can be significantly advanced, offering a nuanced strategy to harness the immune system’s power against cancer. Figure 3 illustrates the immune response model involving KIF20A. 

In the realm of cancer immunotherapy, the methodology for antigen administration plays a pivotal role in leveraging DCs’ antigen-processing capabilities to elicit an effective immune response. These methods encompass a variety of approaches including intravenous, intramuscular, subcutaneous, and intracutaneous routes, each significant for the introduction of antigens into DCs for subsequent immune activation. The efficacy of vaccine strategies is contingent upon multiple factors such as the type of antigen (whether MHC-I or MHC-II restricted), the dosage, the choice of adjuvant, and the route of administration. Notably, DNA, RNA, and synthetic long peptides (SLPs) have been identified as promising in generating robust T cell responses, offering potential avenues for cancer vaccine development [79,80,81].

DNA vaccines and RNA vaccines differ in their manufacturing processes and the way they induce immune responses, with both showing recent clinical effectiveness. DNA vaccines have demonstrated strong CD8+ T cell responses, particularly in targeting HPV-related precancerous lesions and in animal models [82], while RNA vaccines have shown promise in the treatment of melanoma [83,84]. Synthetic long peptide (SLP) vaccines represent an innovative approach in cancer immunotherapy by targeting tumor antigens, activating both CD8+ and CD4+ T cells to induce a robust immune response. SLP vaccines have shown clinical potential in targeting a wide range of cancers [85,86,87].

Non-antigen-specific in situ vaccines (ISVs) represent an alternative strategy, activating immune responses through various mechanisms without necessitating the identification of specific tumor antigens [88]. These include agents that stimulate innate immune pattern recognition receptors (PRRs), oncolytic viruses, and immune-activating cytokines. Such approaches have shown potential in inducing both local and systemic antitumor responses, expanding the toolkit available for cancer immunotherapy and offering novel pathways to combat tumor growth and proliferation.

In the landscape of cancer immunotherapy, KIF20A has emerged as a prominent target across various studies, indicating its pivotal role as a novel TAA with significant therapeutic potential. The foundational work by K Imai et al. was pivotal in identifying KIF20A’s overexpression in pancreatic cancer, setting the stage for subsequent immunotherapy research by demonstrating the ability of specific peptides to induce HLA-A2-restricted CD8+ T-cells targeting KIF20A(+) HLA-A2(+) tumor cells, marking a notable advancement in the field [89]. This discovery laid the groundwork for further exploration into the immunotherapeutic utility of KIF20A, as evidenced by Osawa et al., who broadened the understanding by identifying HLA-A*2402-restricted epitope peptides from KIF20A, further underscoring its significance in pancreatic and stomach cancers [90].

The work by Yusuke Tomita et al. brought to light the capacity of KIF20A-derived long peptides to invoke both CD4+ T-cell and CD8+ T-cell responses, showcasing the versatility of KIF20A-LPs in immunotherapy, especially for head-and-neck malignant tumors (HNMT), offering a glimmer of hope for patients with these challenging conditions [91]. Clinical trials have been instrumental in translating these findings into potential treatments. For instance, Shingo Asahara et al. demonstrated the significant efficacy and safety of a KIF20A-66 peptide vaccine in advanced pancreatic cancer, pointing towards a paradigm shift in treating this lethal disease [92]. Similarly, the VENUS-PC study by Nobuaki Suzuki et al. evaluated a peptide cocktail therapy, including KIF20A, revealing a safe profile and suggesting improved patient outcomes, thereby reinforcing the vaccine’s potential in clinical settings [93].

The initiative by Junji Yatsuda et al. to employ HLA-DR4 transgenic mice for identifying CD4+ T cell epitopes among TAAs like KIF20A exemplifies the innovative approaches being undertaken to screen candidate peptides for vaccine development, integrating experimental research with computational algorithms [94]. The studies by Ryuji Okuyama et al. and Atsushi Aruga et al., demonstrating the tolerability and immunogenicity of KIF20A-targeting peptide vaccines, indicate a promising avenue for stable disease management in advanced pancreatic and biliary tract cancers [95,96]. This evidence suggests a broader applicability of KIF20A in immunotherapy beyond pancreatic cancer, hinting at a multi-cancer target potential.

The research by Motoki Miyazawa et al. and Yoshiyuki Fujiwara et al., focusing on postoperative adjuvant treatment with KIF20A-containing peptide vaccines, has shown promising safety and immunogenicity, with significant survival benefits linked to KIF20A-specific CD8+ T-cell responses, offering a beacon of hope for surgically resected pancreatic cancer patients [97,98]. Furthermore, pilot studies, such as those conducted by Fujiwara et al. and Ryogo Kikuchi et al., evaluating KIF20A-targeting vaccines in conjunction with chemotherapy for advanced gastric cancer and high-grade glioma, respectively, have demonstrated manageable safety profiles and suggested potential therapeutic efficacy [99,100].

The Phase II trial by Mutsunori Murahashi et al., targeting KIF20A in advanced biliary tract cancer, has further expanded the horizon, illustrating the vaccine’s clinical benefits through induced specific T-cell responses and prolonged survival [101]. Importantly, the systematic analysis by Paul Schossig et al. has positioned KIF20A alongside other TAAs like CT45 and LY6K as promising candidates for immunotherapy in epithelial ovarian cancer, based on high expression levels and prognostic relevance, emphasizing the antigen’s broad therapeutic implications [102].

Crucially, the study by Yu Akazawa et al. evaluated the NCCV Cocktail-1 vaccine, containing peptides from KOC1, FOXM1, and KIF20A, in pediatric refractory solid tumors, showcasing its safety and potential efficacy, with observed stable disease in some patients and a correlation between higher peptide-specific CD8+ T-cell frequencies and better progression-free survival [103]. This highlights the versatility of KIF20A-targeted immunotherapy across different age groups and tumor types, underscoring its potential as a universal cancer therapy target.

In the realm of cholangiocarcinoma, Akihiko Kida et al.’s investigation into immune responses against TAA-derived CD8+ T-cell epitopes identified KIF20A among other TAAs as promising targets for immunotherapy. The study elucidated that epitopes from KIF20A and other TAAs elicited significant immune responses in patients, suggesting their utility in devising immunotherapeutic strategies for this challenging cancer type. Notably, the correlation between higher lymphocyte counts and responses to multiple TAA-specific CD8+ T-cells with prolonged overall survival accentuates the critical role of KIF20A as a target for immunotherapy, thereby enriching the immunotherapeutic arsenal against cholangiocarcinoma and potentially other cancers [104].

The collective insights from these studies accentuate the pivotal role of KIF20A as a versatile and effective target in cancer immunotherapy. The journey from its initial discovery to its application in various clinical trials underscores the immense potential of KIF20A-targeted therapies. The ability of KIF20A-derived peptides to elicit robust immune responses across different cancer types, coupled with their tolerability and potential for improved patient outcomes, paves the way for a new era of cancer treatment. Moreover, the systematic approach to TAA prioritization, as demonstrated by Schossig et al., further refines the selection process for potential immunotherapy targets, ensuring that the most promising candidates, like KIF20A, are brought to the forefront of cancer research and treatment.

This burgeoning body of evidence underlines the necessity for continued exploration of KIF20A in cancer immunotherapy. Future large-scale, randomized clinical trials are imperative to unequivocally establish the efficacy and safety of KIF20A-targeted therapies. Additionally, the ongoing research must also focus on identifying predictive biomarkers for patient selection, optimizing vaccine formulations, and combining KIF20A-targeted therapies with other immunotherapeutic modalities to enhance treatment efficacy and durability. The promising outcomes observed in trials thus far offer a glimpse into the potential of KIF20A as a cornerstone in the fight against cancer, heralding a new chapter in the quest for more effective, targeted, and personalized cancer therapies. Figure 4 and Table 3 illustrate a comprehensive overview of the findings and data accumulated from the research.

## 7. Challenges and Future Directions in KIF20A-Targeted Cancer Therapy

### 7.1. Selectivity and Specificity

One of the primary challenges in targeting KIF20A is ensuring selectivity and minimizing off-target effects. KIF20A plays a crucial role in normal cellular division and transport processes, so targeting this kinesin in cancer cells without affecting its function in healthy cells is critical to avoid unwanted side effects [105].

### 7.2. Resistance Mechanisms

Cancer cells are notoriously adept at developing resistance to targeted therapies [106,107]. Understanding the mechanisms by which tumors may evade KIF20A inhibition is essential for the development of durable treatments. This includes the potential for cancer cells to activate compensatory pathways or mutations in KIF20A that reduce the efficacy of targeted agents.

### 7.3. Heterogeneity of Tumor Response

Tumors can exhibit significant biological heterogeneity, both between different types of cancer and within a single tumor [108,109]. This diversity can affect the expression levels of KIF20A and, consequently, the effectiveness of KIF20A-targeted therapies across and within tumor types.

### 7.4. Delivery Challenges

Efficiently delivering KIF20A-targeted therapies, especially in the context of immunotherapy such as peptide vaccines, poses significant challenges [110,111]. Enhancing the stability of these agents, ensuring targeted delivery to tumor cells, and effectively presenting antigens to the immune system are critical hurdles to overcome. Nanotechnology and other novel delivery systems present opportunities to improve the delivery and efficacy of KIF20A-targeted therapies. Targeted delivery systems can enhance the concentration of therapeutic agents at the tumor site while minimizing systemic exposure.

### 7.5. Precision Medicine Approaches

Advances in genomics and proteomics offer opportunities to identify patients most likely to benefit from KIF20A-targeted therapies based on the molecular characteristics of their tumors. Personalized treatment regimens can be developed by integrating KIF20A expression levels, genetic mutations, and signaling pathway activities.

### 7.6. Combination Therapies

Combining KIF20A-targeted agents with other therapeutic modalities, such as chemotherapy, radiation, or immune checkpoint inhibitors, may enhance efficacy and overcome resistance. Synergistic combinations can be identified through preclinical studies and clinical trials.

### 7.7. Immune System Modulation

Further research into how KIF20A-targeted therapies can be optimized to modulate the immune response against tumors will be crucial. This includes exploring adjuvants that can enhance the immunogenicity of KIF20A peptides and strategies to overcome tumor-induced immunosuppression.

### 7.8. Comprehensive Understanding of KIF20A Function

Deeper insights into the biology of KIF20A, including its role in cancer progression and interaction with other cellular proteins, will inform the development of more effective therapeutic strategies. Advanced computational models and high-throughput screening methods can accelerate this understanding.

As we continue to explore the therapeutic potential of targeting KIF20A in cancer, overcoming these challenges and leveraging new technologies and insights will be critical. The future of KIF20A-targeted cancer therapy lies in the intersection of innovative research, collaborative efforts across disciplines, and a commitment to translating scientific discoveries into clinical advances. With ongoing research and clinical development, KIF20A stands as a beacon of hope for improving cancer treatment outcomes and offering patients a brighter future.

## 8. Conclusions

The exploration of KIF20A within the context of cancer biology and therapy presents a promising frontier in the ongoing battle against cancer. The detailed examination of KIF20A’s expression, structure, and multifaceted roles in tumor progression, as discussed in this review, underscores its significance not only as a marker for cancer prognosis but also as a viable target for innovative therapeutic strategies. The emerging evidence supporting the efficacy of KIF20A-targeted immunotherapies, particularly peptide vaccines, heralds a new era in the personalized treatment of cancer, offering hope for improved patient outcomes.

However, the path to integrating KIF20A-targeted therapies into clinical practice is fraught with challenges, including but not limited to ensuring treatment specificity, combating resistance mechanisms, and optimizing delivery methods. These obstacles necessitate a multifaceted approach, combining the latest advances in precision medicine, combination therapies, and novel drug delivery systems. Furthermore, a deeper understanding of KIF20A’s biological functions and its interplay with the tumor microenvironment will be crucial for developing more effective and durable cancer treatments.

## Figures and Tables

**Figure 1 cancers-16-02958-f001:**
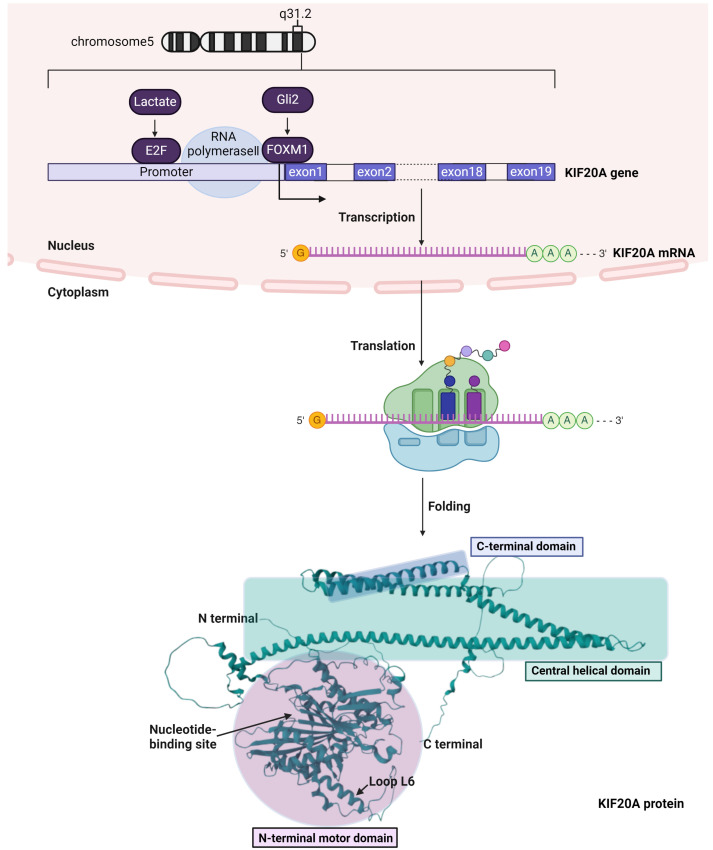
Regulation and Functional Architecture of KIF20A. At the chromosomal level, KIF20A is localized on chromosome 5 at q31.2. Regulatory elements such as Gli2 and FOXM1 are depicted; these factors enhance transcriptional activity at the promoter region through interactions with the forkhead responsive element (FHRE). Environmental factors such as lactate are shown modulating gene expression by influencing transcription factors, including E2F. The resultant KIF20A mRNA is processed within the nucleus and subsequently transported into the cytoplasm for translation. In the cytoplasm, the KIF20A mRNA is translated into the protein, which then folds into its active form. The 3D structure of the KIF20A protein is segmented into three distinct domains: the N-terminal motor domain, the central helical domain, and the C-terminal domain. The nucleotide-binding site (NBS), crucial for the protein’s ATPase activity, is prominently marked within the motor domain. Additionally, the loop L6, which is considerably longer than comparable loops in other kinesins, is accentuated to underscore its structural and functional importance. Cartoon in Figure 1 was created with https://BioRender.com and accessed on 1 April 2024.

**Figure 2 cancers-16-02958-f002:**
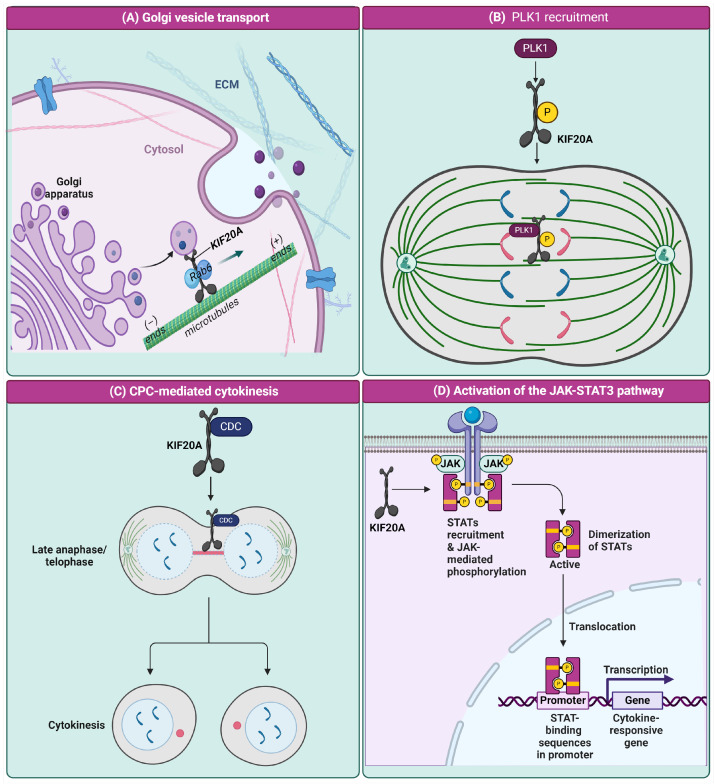
Biological functions of KIF20A. (**A**) Golgi vesicle transport: KIF20A mediates the transport of vesicles from the Golgi apparatus towards the extracellular matrix (ECM) via microtubules. The main panel shows the Golgi apparatus with KIF20A guiding RAB6-positive vesicles along microtubules towards their destinations within the cell. Inset diagrams highlight the molecular interactions critical for this process: one shows the N-terminal motor domain of KIF20A interacting with tubulin, which is essential for vesicle propulsion along microtubules; the other depicts the central helical domain of KIF20A in complex with RAB6, demonstrating the specific binding that facilitates precise vesicle routing. (**B**) PLK1 recruitment: The phosphorylation of KIF20A by PLK1 is shown, which is essential for the recruitment of PLK1 to the central spindle during mitosis, facilitating proper chromosome segregation. (**C**) CPC-mediated cytokinesis: In late anaphase/telophase, KIF20A is shown interacting with the CPC at the central spindle, which is critical for the completion of cytokinesis and the physical separation of the daughter cells. (**D**) Activation of the JAK/STAT3 pathway: The illustration depicts the role of KIF20A in activating the JAK/STAT3 pathway. The phosphorylation of STAT3 leads to its dimerization, translocation to the nucleus, and the transcription of genes that contribute to the development of drug resistance. Cartoon in Figure 2 was created with https://BioRender.com and accessed on 3 April 2024.

**Figure 3 cancers-16-02958-f003:**
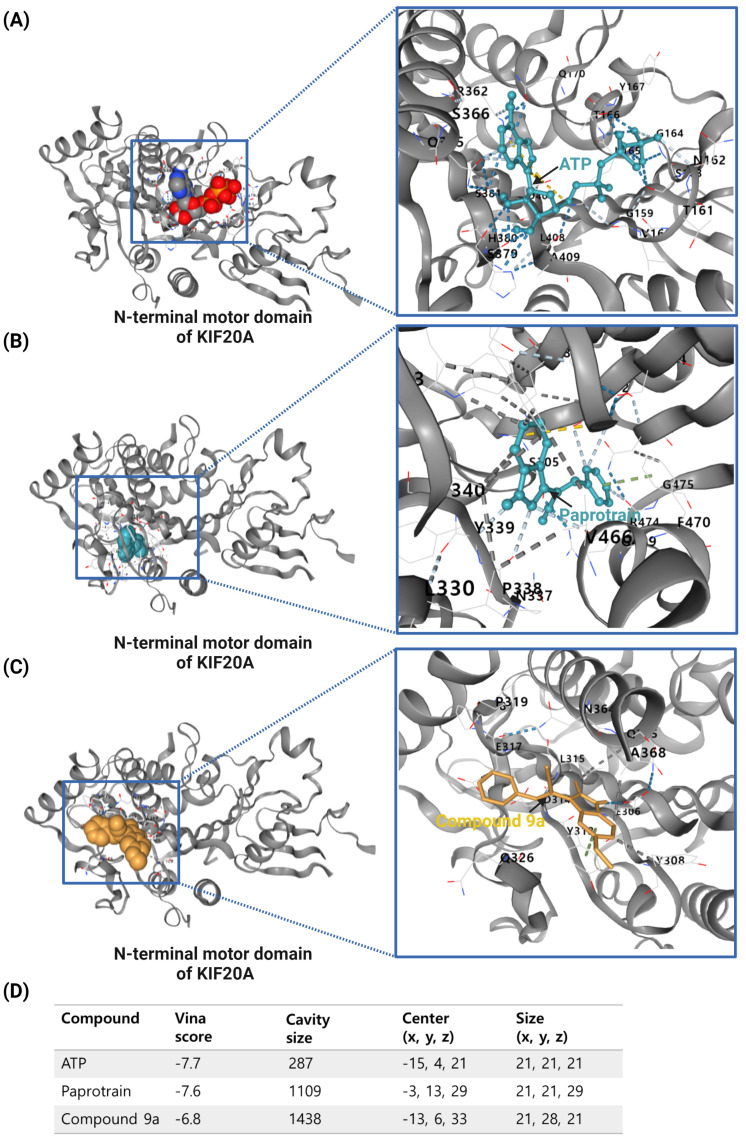
Comparative Docking Analysis of ATP, Paprotrain, and Compound 9a on the N-terminal Motor Domain of KIF20A. (**A**) Illustration of ATP (red spheres) bound to the N-terminal motor domain of KIF20A, highlighting the key residues involved in the binding interaction. The inset provides a closer view of the ATP binding site, showing interactions with surrounding amino acids. (**B**) Visualization of Paprotrain (cyan sticks) within the binding cavity of the KIF20A motor domain. The inset zooms into the interaction details, depicting how Paprotrain aligns and forms contacts with critical residues such as Tyr339, Pro338, and Arg474. (**C**) Display of Compound 9a (orange sticks) docked at the motor domain of KIF20A. The inset focuses on the binding interactions of Compound 9a, showcasing its engagement with residues like Glu306 and Tyr313, and the expanded binding cavity compared to Paprotrain. The blue lines in the attached file are used to highlight specific regions within the 3D structures of the N-terminal motor domain of the KIF20A protein. These lines connect the overall structure on the left side of each subfigure (**A**–**C**) to a zoomed-in view on the right, which details the interactions between the KIF20A protein and the compounds ATP, Paprotrain, and Compound 9a. (**D**) Table summarizing the Vina docking scores, cavity sizes, center coordinates, and dimensions for ATP, Paprotrain, and Compound 9a. This table provides a quantitative comparison of how each compound interacts with the motor domain of KIF20A, underscoring differences in binding affinity and cavity adaptation. The docking analysis was conducted using CB-Dock, and the downloaded data were modified on https://BioRender.com and accessed on 10 April 2024.

**Figure 4 cancers-16-02958-f004:**
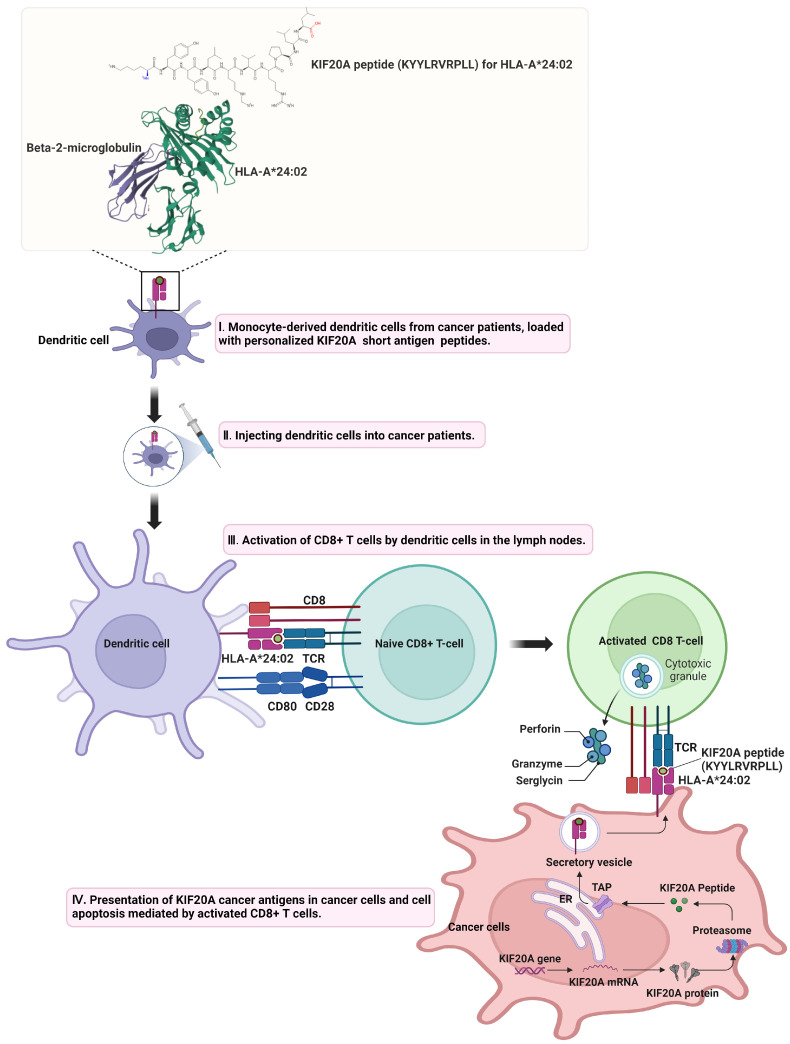
KIF20A Peptide-Driven Immune Activation: Pathways to Cancer Vaccine Development. The expanded diagram would elaborate on the immune response mechanism initiated by KIF20A peptides, starting from the moment the peptides are introduced into the body. It would trace the journey of these peptides as they are ingested by dendritic cells, highlighting the cells’ migration to the lymph nodes. HLA-A24 (A*24:02) binds to MHC class I and activates CD8+ T cells. The * symbol separates the gene locus from the allele designation. In the diagram at the top of the image, the red carboxyl group and the blue amino group represent the key functional groups interacting with HLA-A24 (A*24:02). Although it is not shown in the figure, KIF20A-LP binds to both MHC class I and II, thereby activating both CD4+ T cells and CD8+ T cells. Activated CD8+ T cells recognize and bind to cancer cells presenting the KIF20A antigen. This interaction triggers the release of cytotoxic granules containing perforin, granzyme, and serglycin, leading to the targeted apoptosis of the cancer cells. Cartoon in Figure 4 was created with https://BioRender.com and accessed on 18 April 2024.

**Table 1 cancers-16-02958-t001:** KIF20A’s expression patterns in tumor and adjacent normal tissues were examined through RNA-Seq and chip data accessed via TNMPlot, utilizing paired samples for the analysis. The data were downloaded from TNMPlot.

RNA-Seq Data				
Tissue	Mann_Whitney.p. ^a^	Fold.Change.Mean ^b^		Fold.Change.Median ^c^
**Blader Urothelial Carcinoma**		7.79 × 10^−4^	3.88	19.00
**Breast invasive Carcinoma**		3.59 × 10^−19^	8.64	13.01
**Cervical Squamous Cell Carcinoma and Endocervical Adenocarcinoma**		1.81 × 10^−1^	38.88	74.08
**Cholangiocarcinoma**		9.15 × 10^−3^	67.07	42.67
**Colon Adenocarcinoma**		2.72 × 10^−6^	2.94	4.27
**Esophageal Carcinoma**		1.41 × 10^−2^	5.10	3.64
**Head and Neck Carcinoma**		1.69 × 10^−6^	2.72	2.79
**Kidney Chromophobe**		2.34 × 10^−2^	3.90	1.38
**Kidney Renal Clear Cell Carcinoma**		1.28 × 10^−11^	5.41	6.18
**Kidney Renal Papillary Cell Carcinoma**		6.56 × 10^−6^	10.90	5.92
**Liver Hepatocellular Carcinoma**		1.26 × 10^−9^	20.78	16.50
**Lung Adenocarcinoma**		6.2 × 10^−11^	11.47	10.19
**Lung Squamous Cell Carcinoma**		1.14 × 10^−9^	13.43	18.73
**Pancreatic Adenocarcinoma**		1 × 10^−1^	2.21	2.02
**Pheochromocytoma and Paraganglioma**		1.81 × 10^−1^	1.59	2.56
**Prostate Adenocarcinoma**		9.67 × 10^−8^	2.84	2.73
**Rectum Adenocarcinoma**		9.15 × 10^−3^	3.75	3.69
**Sarcoma**		1 × 10^0^	10.65	10.65
**Gene chip data**				
**Breast**		9.16 × 10^−6^	2.00	1.93
**CNS**		1.81 × 10^−1^	11.05	1.17
**Colon**		1.16 × 10^−23^	2.32	2.64
**Gastric**		4.78 × 10^−38^	2.57	2.59
**Kidney**		2.9 × 10^−32^	5.22	6.87
**Liver**		3.87 × 10^−21^	5.08	7.70
**Lung**		9.62 × 10^−41^	4.26	6.16
**Lymphoid**		3 × 10^−2^	0.39	0.26
**Neural**		1.81 × 10^−1^	1.74	1.62
**Esophageal**		2.52 × 10^−8^	4.85	4.91
**Oral cavity**		1 × 10^−1^	2.20	2.33
**Ovarian**		5.91 × 10^−2^	3.41	4.96
**Pancreas**		1.58 × 10^−8^	2.61	2.58
**Prostate**		1.71 × 10^−2^	1.74	1.57
**Skin**		3.34 × 10^−3^	2.22	3.19
**Soft tissue**		2.01 × 10^−1^	1.29	1.35
**Thyroid**		2.01 × 10^−1^	1.29	1.35
**Uterus**		2.01 × 10^−1^	1.29	1.35

a: Reflects the *p*-value from the Mann–Whitney U test, a nonparametric method assessing significant differences between two distinct groups, utilized here to compare KIF20A gene expression in normal versus tumor tissues. b: Denotes the mean fold change in expression of the KIF20A gene, comparing tumor to normal tissues, with fold change signifying the ratio of expression alteration between two given states, typically normal and disease conditions in genetic research. c: Similar to mean fold change, this indicates the median fold change for KIF20A expression, representing the middle value in an ordered data set and offering a central tendency measure that is more robust to extreme values and distribution skewness.

**Table 2 cancers-16-02958-t002:** Role of inhibitors of KIF20A in cancer.

Inhibitor Type	Model (Cell or Animal)	Inhibitor Concentration	Results	Ref.
Paprotrain 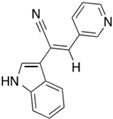	HeLa Cells	10 to 50 μM	Increased binucleated cells, disrupted chromosome passenger protein relocation, specific inhibition of KIF20A without affecting Kif4 or MKLP-1	[54]
Paprotrain	HeLa Cells	50 μM	Reduced motor activity and transport efficiency of KIF20A, decreased number and velocity of coreCPC complexes, and inhibited recruitment of coreCPC	[55]
Paprotrain	Castration-resistant prostate cancer cells	500 nM	Reduced castration-resistant proliferation by inhibiting AR signaling activation driven by KIF20A.	[56]
Paprotrain	Bladder cancer cell lines (RT112, T24, UMUC3)	0 to 20 μM	Combined with the drug palbociclib, it significantly enhances the effectiveness of the treatment, leading to a more substantial reduction in cancer cell survival.	[57]
Paprotrain	HeLa cells with fluorescently tagged histones	0 to 11 μM	Induced significant mitotic arrest by disrupting chromosome congression, increased Aurora kinase activity, and led to chromosomal instability.	[58]
Compound 9a 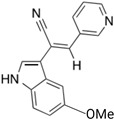	Human tumor cell lines (e.g., MIA-PaCa-2, HCT116)	0 to 50 μM	Demonstrated 2- to 10-fold higher potency than Paprotrain in inhibiting KIF20A ATPase activities; significant increase in antiproliferative effects across different tumor cell lines.	[59]
BKS0349 (unpublished)	Xenograft mouse model (endometriosis)	200 mg/kg/week	Significantly reduced size of endometriotic lesions, decreased cell proliferation, increased apoptosis.	[60]

**Table 3 cancers-16-02958-t003:** Overview of KIF20A-Targeted Immunotherapies in Cancer Research and Clinical Trials.

Products	Clinical Stage	Vaccine Targeting	Main Results	Ref.
KIF20A-2, KIF20A-8, KIF20A-28 peptides	Preclinical	KIF20A(+) HLA-A2(+) pancreatic cancer cells	Identified peptides induce HLA-A2-restricted CD8+ T-cells in transgenic mice and human CD8+ T-cells in vitro, targeting pancreatic cancer cells expressing KIF20A and HLA-A2 without causing autoimmunity.	[89]
KIF20A-10-66 peptide	Preclinical	HLA-A*2402+ cancer cells expressing KIF20A	KIF20A-10-66 peptide induces specific CD8+ T-cells that target and exhibit cytotoxic activity against cancer cells expressing KIF20A and HLA-A*2402, demonstrating potential for CD8+ T-cell-inducing cancer therapies.	[90]
KIF20A long peptides (KIF20A-LPs)	Preclinical	KIF20A+ HNMT expressing HLA-A2 or HLA-A24	Identified KIF20A-LPs induce KIF20A-specific TH1 and CD8+ T-cell responses in vivo and in vitro, with significant TH1 responses in 50% of HNMT patients, associating with KIF20A expression in tumor tissues.	[91]
KIF20A-66 peptide	Phase I/II	Advanced pancreatic cancer	The vaccine was well tolerated, showed a disease control rate of 72%, and significantly prolonged survival compared to historical controls, indicating effective immunotherapy against advanced pancreatic cancer.	[92]
KIF20A peptide, VEGFR1 peptide, VEGFR2 peptide	Phase II	Advanced pancreatic cancer	No severe adverse effects observed; patients with peptide-specific CD8+ T-cell induction for KIF20A or VEGFR1 showed better prognosis; therapeutic peptide cocktail may be effective in patients with peptide-specific immune reactions.	[93]
KIF20A494-517 peptide	Preclinical	HLA-DR4 transgenic mice	Successful induction of CD4+ T cell responses to KIF20A and other TAAs in transgenic mice and HLA-DR4-positive human PBMCs, validating the approach for vaccine peptide screening.	[94]
KIF20A (KVYLRVRPLL) peptide	Phase I	Advanced pancreatic cancer	The KIF20A component of the multi-peptide vaccine was well-tolerated and induced specific T-cell responses, contributing to observed clinical benefits in patients.	[95]
KIF20A peptide	Phase I	Advanced biliary tract cancer	The vaccine was well-tolerated, induced peptide-specific T-cell responses, and stable disease was observed in 5 of 9 patients. Median PFS and OS were 3.4 and 9.7 months, respectively.	[96]
KIF20A peptide, VEGFR1 peptide, VEGFR2 peptide	Phase II	Resected pancreatic cancer	The vaccine was well-tolerated; median DFS was 15.8 months. Patients with KIF20A-specific CD8+ T-cell responses and/or KIF20A expression showed better DFS.	[97]
Peptide cocktail vaccine (including KIF20A peptide)	Phase II	Surgically resected pancreatic cancer patients	The vaccine was safe and induced specific immune responses. Patients showing KIF20A-specific CD8+ T-cell responses had significantly better disease-free survival, highlighting its potential as an effective postoperative adjuvant treatment.	[98]
KIF20A peptide (part of a cocktail vaccine)	Pilot study (Post-operative adjuvant)	Advanced gastric cancer	The combination of vaccine therapy and S-1 was safe and manageable as adjuvant therapy for stage III gastric cancer, achieving optimal relative dose intensity of S-1 with manageable injection-site reactions.	[99]
LY6K, DEPDC1, KIF20A, FOXM1, VEGFR1, VEGFR2 peptides	Pilot Study	Recurrent/progressive high-grade glioma (HGG)	Well-tolerated treatment inducing robust T-lymphocyte responses, with a median OS of 9.2 months and progression-free status in some patients for at least six months.	[100]
OCV-C01 (VEGFR1, VEGFR2, KIF20A peptides)	Phase II	Advanced biliary tract cancer	Induced vaccine-specific T-cell responses in patients, with observed contributions to prolonged overall survival.	[101]
KIF20A	Target Selection Study	Epithelial ovarian cancer	KIF20A identified as a high-priority antigen for T cell therapy due to its expression and prognostic relevance in EOC, supported by immunohistochemical staining and HLA-ligandome analysis.	[102]
NCCV Cocktail-1 (KOC1, FOXM1, KIF20A peptides)	Phase I	Pediatric refractory solid tumors	Well tolerated; induced peptide-specific CD8+ T-cells for KOC1, FOXM1, and KIF20A; associated with better progression-free survival in patients with high CD8+ T-cell frequencies.	[103]
KIF20A-derived epitopes	Preclinical	Cholangiocarcinoma	Identified KIF20A among epitopes stimulating specific immune responses in cholangiocarcinoma patients, with potential for immunotherapy. Higher lymphocyte counts correlated with TAA-specific response, and overall survival was significantly prolonged in patients with two or more TAA-specific CD8+ T-cell responses.	[104]

## Data Availability

The data presented in this study are available on request from the corresponding author due to (specify the reason for the restriction).

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
