# Peer review of "Advancing Cancer Therapy: The Role of KIF20A as a Target for Inhibitor Development and Immunotherapy"

_cancers, 2024, doi:10.3390/cancers16172958_

Round 1

Reviewer 1 Report

Comments and Suggestions for Authors

Dong Oh Moon's review focuses on recent advances in the development of inhibitors and the use of kinesin family member 20A (KIF20A) as a target for cancer immunotherapy.

KIF20A recognized as for its significant role in cancer and is highly pressed in various type of cancer. Given its crucial role in cancer progression, KIF20A is a promising target for therapeutic intervention and valuable prognostic marker for various malignancies.

The concept of the manuscript is interesting and worthy of publication. However, the manuscript needs to be revised before it can be accepted for publication. These are some suggestions/questions that can help the authors improve the manuscript.

-          Line 317 ‘This pathway, when activated, can lead to the transcription of genes that promote cancer cell proliferation’

Starting of this, it is necessary to explain:

What changes precede or indicate this promotion? What type of activation does this do? This should be discuss or proposed.

Because in vivo, the JAK-STAT3 pathway signalling mechanism is involved in various cellular processes, including immune response, cell growth and differentiation. These pathways are activated by cytokines and growth factors regulating gene expression in response to extracellular signals.

-          Lines 410–412 simply duplicate sentence ‘line 407–409’. This part should be changed.

-          In part 5 of review regarding inhibiting of KIF20A, the inhibitors could disrupt of normal cell division processes, potentially leading to adverse effect in rapidly dividing normal cell, such as those in gastrointestinal tract or bone marrow.

-          Can the author add some information regarding adverse effect of KIF20A inhibition?

-          Requires minor editing in English.

Comments on the Quality of English Language

-          Requires minor editing in English.

Author Response

  Line 317 ‘This pathway, when activated, can lead to the transcription of genes that promote cancer cell proliferation’

Starting of this, it is necessary to explain:

What changes precede or indicate this promotion? What type of activation does this do? This should be discuss or proposed.

Because in vivo, the JAK-STAT3 pathway signalling mechanism is involved in various cellular processes, including immune response, cell growth and differentiation. These pathways are activated by cytokines and growth factors regulating gene expression in response to extracellular signals.

⟶ I have made revisions based on your feedback.

-          Lines 410–412 simply duplicate sentence ‘line 407–409’. This part should be changed.

⟶ I have updated the document to incorporate your review suggestions.

-          In part 5 of review regarding inhibiting of KIF20A, the inhibitors could disrupt of normal cell division processes, potentially leading to adverse effect in rapidly dividing normal cell, such as those in gastrointestinal tract or bone marrow.

  Can the author add some information regarding adverse effect of KIF20A inhibition?

⟶ Thank you for your critical review. The following information has been additionally written and added. “The study of KIF20A inhibition and its effects on normal tissues remains insufficiently explored. Recent findings indicate that KIF20A suppression leads to early cell cycle exit and accelerated neuronal differentiation in both normal cerebellar granule neuron progenitors (GNPs) and tumor-initiating GNPs. These results suggest that KIF20A plays a crucial role not only in tumor progression but also in the normal development of cerebellar neurons [34]. Furthermore, an analysis of breast cancer subtypes revealed that KIF20A was expressed in 195 (75.9%) of tumor samples, while it was rarely detected in normal breast tissues [29]. This indicates that KIF20A inhibition may have minimal impact on certain normal tissues. Nevertheless, comprehensive studies are required to better understand the broader implications of KIF20A inhibition on normal cells, as well as to determine its therapeutic potential across different cancer types. Further research is essential to evaluate the selective targeting of KIF20A in cancer treatment while minimizing adverse effects on normal tissues.”

-          Requires minor editing in English.

⟶ Some sentences have been corrected.

Reviewer 2 Report

Comments and Suggestions for Authors

Recent increase in publications on KIF20A shows its importance as a target of cancer therapy. The author intends to provide an integrated review on KIF20A describing its expression and protein structure, its expression in cancers, its multiple biological functions, KIF20A inhibitors, and KIF20A-targeted cancer immunotherapy. Although the MS could be more compact, the description is clear and easy to understand. The references in recent years are properly cited.

Author Response

Recent increase in publications on KIF20A shows its importance as a target of cancer therapy. The author intends to provide an integrated review on KIF20A describing its expression and protein structure, its expression in cancers, its multiple biological functions, KIF20A inhibitors, and KIF20A-targeted cancer immunotherapy. Although the MS could be more compact, the description is clear and easy to understand. The references in recent years are properly cited.

⟶ Thank you very much for your review.

Reviewer 3 Report

Comments and Suggestions for Authors

The manuscript offers a thorough review of KIF20A, emphasizing its potential as a therapeutic target in cancer treatment. The detailed discussion of its molecular structure, expression patterns, and roles in various cancers provides a solid foundation for understanding the significance of KIF20A in oncology. The examination of KIF20A as a target for inhibitor development and immunotherapy is both timely and relevant, particularly in the context of the current focus on precision medicine and targeted therapies. However, several areas need to be addressed to enhance the overall quality of the manuscript:

  1. The phrase "Recruitment of" is duplicated on line 211 and should be corrected.
  2. Several statements require references to support the claims made, particularly on lines 217-218, 259-261, and 268-276.
  3. A discussion on the safety profile of KIF20A inhibitors is necessary, along with a comparison of the efficacy of various inhibitors.
  4. The content in lines 556-596 appears to be loosely related to KIF20A and may not be essential for this manuscript. Consider reducing or removing this section to maintain focus on KIF20A.
Comments on the Quality of English Language

The English Language is fine.

Author Response

The manuscript offers a thorough review of KIF20A, emphasizing its potential as a therapeutic target in cancer treatment. The detailed discussion of its molecular structure, expression patterns, and roles in various cancers provides a solid foundation for understanding the significance of KIF20A in oncology. The examination of KIF20A as a target for inhibitor development and immunotherapy is both timely and relevant, particularly in the context of the current focus on precision medicine and targeted therapies. However, several areas need to be addressed to enhance the overall quality of the manuscript:

  1. The phrase "Recruitment of" is duplicated on line 211 and should be corrected.

⟶ I have made revisions based on your feedback.

  1. Several statements require references to support the claims made, particularly on lines 217-218, 259-261, and 268-276.

⟶ Edited by adding reference.

  1. A discussion on the safety profile of KIF20A inhibitors is necessary, along with a comparison of the efficacy of various inhibitors.

⟶ Thank you for your critical review. The following information has been additionally written and added. “The study of KIF20A inhibition and its effects on normal tissues remains insufficiently explored. Recent findings indicate that KIF20A suppression leads to early cell cycle exit and accelerated neuronal differentiation in both normal cerebellar granule neuron progenitors (GNPs) and tumor-initiating GNPs. These results suggest that KIF20A plays a crucial role not only in tumor progression but also in the normal development of cerebellar neurons [34]. Furthermore, an analysis of breast cancer subtypes revealed that KIF20A was expressed in 195 (75.9%) of tumor samples, while it was rarely detected in normal breast tissues [29]. This indicates that KIF20A inhibition may have minimal impact on certain normal tissues. Nevertheless, comprehensive studies are required to better understand the broader implications of KIF20A inhibition on normal cells, as well as to determine its therapeutic potential across different cancer types. Further research is essential to evaluate the selective targeting of KIF20A in cancer treatment while minimizing adverse effects on normal tissues.”

  1. The content in lines 556-596 appears to be loosely related to KIF20A and may not be essential for this manuscript. Consider reducing or removing this section to maintain focus on KIF20A.

⟶ I have made revisions based on your feedback.

Round 2

Reviewer 3 Report

Comments and Suggestions for Authors

The author has properly addressed my previous comments.